**Data Availability Statement:** Data cannot cannot be shared publicly because such sharing of individual participant data was not included in the

# The influence of spouses and their driving roles in self-regulation: A qualitative exploration of driving reduction and cessation practices amongst married older adults

**Boon Hong Ang**[1], **Jennifer Anne Oxley**[2], **Won Sun Chen**[3]*, **Michelle Khai Khun Yap**[1], **Keang Peng Song**[1], **Shaun Wen Huey Lee**[4,5,6]*

1 School of Science, Monash University Malaysia, Subang Jaya, Malaysia, 2 Monash University Accident Research Centre, Clayton, Australia, 3 School of Health Science, Swinburne University of Technology, Melbourne, Australia, 4 School of Pharmacy, Monash University Malaysia, Subang Jaya, Malaysia, 5 Gerontology Laboratory, Global Asia in the 21st Century (GA21) Platform, Monash University Malaysia, Subang Jaya, Malaysia, 6 School of Pharmacy, Taylor's University, Subang Jaya, Malaysia

* wchen@swin.edu.au (WSC); shaun.lee@monash.edu (SWHL)

# Abstract

## Introduction

There is growing evidence to suggest the importance of self-regulatory practices amongst older adults to sustain mobility. However, the decision to self-regulate driving is a complex interplay between an individual's preference and the influence of their social networks including spouse. To our best knowledge, the influence of an older adult's spouse on their decisions during driving transition has not been explored.

## Materials and methods

This qualitative descriptive study was conducted amongst married older adults aged 60 years and above. All interview responses were transcribed verbatim and examined using thematic approach and interpretative description method.

## Results

A total of 11 married couples were interviewed. Three major themes emerged: [1] Our roles in driving; [2] Challenges to continue driving; and, [3] Our driving strategies to ensure continued driving. Older couples adopted driving strategies and regulated their driving patterns to ensure they continued to drive safely. Male partners often took the active driving role as the principal drivers, while the females adopted a more passive role, including being the passenger to accompany the principal drivers or becoming the co-driver to help in navigation. Other coping strategies include sharing the driving duties as well as using public transportation or mixed mode transportation.

informed consent of the study. All proposals requesting data access can be directed to the Monash Research Ethics Committee (contact via researchoffice@monash.edu).

**Funding:** The author(s) received no specific funding for this work.

**Competing interests:** The authors have declared that no competing interests exist.

## Discussion

Our findings suggest spouse play a significant role in their partners' decision to self-regulate driving. This underscores a need to recognise the importance of interdependency amongst couples and its impact on their driving decisions and outcomes.

## Introduction

With the projected rise in ageing population worldwide, ensuring safe mobility amongst older adults is a growing concern [1]. Older adults often rely on driving to accomplish their transportation needs. However, as an individual begins to age, his/her skill and ability to drive safely begin to decline [2–4]. To cope with this, older adults often choose to self-regulate their driving patterns (e.g., reduction in terms of speed, distance, and frequency of travel and avoid driving in challenging situations); or in some cases cease driving completely, due to chronic medical conditions or a major life event (e.g., road crash) [5]. According to Ang and associates, self-regulation is a decision-making process that progressively changes over time to compensate for declining abilities or to reduce discomfort experienced in challenging situations by making appropriate adjustments in driving behaviour and alterations to driving pattern [2]. Whilst driving cessation is a major life changing event for many older adults, most of them often try to continue driving safely by engaging in self-regulatory practices [5].

Previous studies have largely focused on exploring an individual's identity with driving, associated factors, and consequences of driving cessation [6–10] which are likely to be important for their social partners. Whilst the decision to self-regulate is often complex and multifactorial, little is known about how and to what extent external factors (e.g., social influence and support) may influence the decision-making process. In most individuals, the process of transitioning from driving to driving cessation is not an individual experience, but often involves a wider social network, including their spouse, children, friends, and even professionals such as clinicians [11,12].

Curl and associates suggest that spouses or partners have a profound impact on the individuals' experiences and their decisions in driving transition, including chauffeuring duty, care giving task, and maintenance of their roles in social engagement [13]. According to Gormley and O'Neill, both married individuals were more likely to be active drivers [14]. Nevertheless, the impact of spousal influence on driving transition amongst older couples is largely understudied. Until recently, most of the available studies have explored spousal influence on life satisfaction and social engagement amongst older couples in developed countries with limited information from developing countries like Malaysia [13, 15,16].

Given that reliance on spouse for transportation support is common amongst older couples [16] and safe travel remains a crucial goal, is there a role for spouse in sustaining safe mobility in later stage of life? Hence, the present study examined the influence of spouses on driving self-regulation. Specifically, this study aims to understand their driving roles and engagements in driving self-regulation and the subsequent impact of their involvement during the transition on driving decisions and outcomes. The findings of the current study may offer new insights into the how spouses influence of spouses on self-regulation and suggest new avenues for programs focused on the roles and contributions of spouse in driving transition and cessation planning in the future.

## Materials and methods

### Participants and definitions

This qualitative study recruited married couples, who were current or former car drivers, aged 60 years and above, and literate in English. Participants were recruited through snowball sampling technique from senior citizen organisations, community centres, and medical centres located in two states in the west coast of Peninsular Malaysia, including Selangor (a more economically developed state) and Kedah (a less economically developed state) [17]. This project was approved by Monash University Human Research Ethics Committee (Project Number: 1517).

The outcome of interest of this study was self-regulation of driving, specifically on reduction and cessation practices. Reduction involves partial avoidance, whereby individuals reduce the use of vehicle, including the distance, frequency of use, and avoid driving in challenging situations such as peak hours, bad weather, and demanding locations [2, 10]. Cessation on the other hand, involves total avoidance, whereby individuals choose to cease driving completely and never drive again [2, 10]. As we aimed to explore the role of spouse in driving self-regulation, we examined three different driving roles that were identified through thematic analyses performed for this study based on the participants' responses (See Theme 1 in the results), defined as follow:

i.  Principal driver: Individual who takes charge of driving duty most of the time.

ii.  Alternate driver: Individual who drives less frequent than the principal driver and takes over driving duty only when the principal driver is not available.

iii.  Former driver: Individual who had ceased driving completely.

### Data collection and instruments

A one-to-one, semi-structured interview was conducted between January 2017 and April 2018 using a list of pre-determined open-ended questions guided by a moderator. To ensure consistency in data collection, the first author conducted all the interviews. The interview guide was developed based upon prior experience of our research team and literature focusing on barriers and facilitators to driving reduction and cessation practices by older married couples to maintain their mobility in later life (S1 File). Couples who consented (written consent) to participate were interviewed separately and received a gift voucher worth RM 50 (approximately USD 11.96) each. Interviews were conducted at a time and location (e.g. a closed room or a quiet place at home, organisation, or medical centre) of convenience for the participants. Each interviews averaged one hour, and responses were audio recorded throughout the sessions. Recruitment of participants was conducted until thematic saturation was achieved when no new themes identified in the subsequent interview. Prior to commencement of the interview sessions, additional demographic information, including medical history and travel patterns were collected using the Driving and Riding Questionnaire (a self-administered battery of existing scales with multiple choice and open-ended questions) [18]. All the interviews were conducted in English and no translation was involved.

### Data analyses

All audio-recorded interviews were transcribed verbatim. The analyses were performed using a thematic approach to identify patterns and emerging themes from the quotes based on

inductive coding technique within the data and not based on pre-existing assumption or theory [19]. In this approach, two coders independently reviewed and familiarised themselves with the interview transcripts and later, collaboratively created a codebook of all preliminary codes that emerged from the independent analyses forming a meaningful framework after mapping of similar codes and renaming of the themes. Peer reviews, including counterchecking with other investigators occurred at several stages of the analyses to ensure rigor and reliability of response coding. Using interpretative description method, the responses from participants were embedded within the text to support and build evidence for the proposed themes and sub-themes. All the quotes included in the text were transcribed exactly from the original recordings. In the event of absence of clarity, additional descriptions were incorporated in parentheses to aid in understanding or [sic] was indicated where necessary. All interviews were coded using NVIVO version 11.0 (QSR International Pty Ltd).

## Results

### Participant characteristics

A total of 11 couples were interviewed with 20 current car drivers and two former car drivers with a median of 68 years old (60–79) (S1 Table). All the participants had some form of education and most were retirees (n = 16, 72.7%). Almost all the current drivers drove their cars daily (n = 15/20, 75.0%), often their own locality (n = 19/20, 95.0%). Most of the participants reported no crash history within the past five years (n = 13, 59.1%). Amongst the two participants who reported to have ceased driving completely, both were females, reported having trouble with their vision, and were living with a spouse who drove a car almost daily.

### Thematic analyses

Three major themes with 11 supporting sub-themes were identified relevant to mobility status: [1] our roles in driving; [2] challenges to continue driving and; [3] our driving strategies to ensure continued driving (Fig 1). All the male partners in this study reported that they were the principal drivers in the household, whilst female partners reported to be either the alternate drivers or former drivers. For many couples who continued to drive, they reported changing their driving habits, such as imposing driving limitations, sharing driving responsibility, and travelling by public transportation or mixed mode transportation (interchanging between public and private transportations for a single trip). For some couples, having a partner for navigation whilst driving was perceived important. More detailed descriptions of the themes with supporting quotes are elucidated in detail below.

**Theme 1: Our roles in driving.** *Sub-theme 1*: *Male partners as principal drivers*. Our analysis revealed that male partners generally took on the principal role of driving the family around, especially in any family trips. Most male participants replied that this was the norm, and they rarely got other family members to drive, unless necessary (e.g., too tired to drive, unfamiliar with roads).

> *"Couple 1 Male: I [take the responsibility to] drive most of the time. In case I feel unwell, my wife will drive. But I drive more [often], and she only drives for her shopping and domestic purposes."*

This need to drive was even stronger amongst males whose female partners had ceased driving as they felt that this was their responsibility to chauffeur their spouse. Only two female participants were former drivers and they heavily relied on their male partners to chauffeur them after ceasing to drive.

| THEME 1 | | THEME 2 | THEME 3 |
|---|---|---|---|
| **OUR ROLES IN DRIVING** | | **CHALLENGES TO CONTINUE DRIVING** | **OUR DRIVING STRATEGIES TO ENSURE CONTINUED DRIVING** |
| Continued driving | Principal Driver **OR** Alternate Driver | **INTERPERSONAL** Spouse condition *(tired, unwell)* **ENVIRONMENT** Travel nature *(long distance trip)* | Share driving |
| | | **DEMOGRAPHIC** Retirement/Change in responsibility/lifestyle **INTRAPERSONAL** Physical and physiological changes | Driving with modification Spouse as a co-driver Spouse as a principal driver |
| Ceased driving | Former Driver | **INTERPERSONAL** Spouse influence **ENVIRONMENT** Road and vehicle conditions | Public transportation Mixed mode transportation |

**Fig 1. Summary of driving roles, obstacles, and strategies adopted by older married couples.**

*"Couple 5 Male: She rely very much on me for transport."*

*"Couple 4 Female: Husband, he is retired already. So, I have got myself a chauffeur."*

*"Couple 5 Female: I have a driver, so no need to drive. My husband drives."*

Our study also noted that it was common for female partners to rely on male partners for transportation. Most of them expressed that they would depend on their spouse for mobility if they were to cease driving someday.

*"Couple 1 Female: If I stopped driving, I will depend on my husband."*

*Sub-theme 2*: *Female partners as alternate drivers*. Our study also found that female spouses were often the alternate driver. Just like principal drivers, alternate drivers held an important driving role in their early phase of driving, to support the family needs, including chauffeuring

the children to school related activities or visiting their grandparents. As they begin to age, most female participants suggest that they continued driving for personal needs.

> *"Couple 3 Female: When my children were small, I have to take them to see doctor, take them to tuition, and take them for activities."*

> *"Couple 9 Female: Every day, I used to drive to work before I retire, drive my children to tuition, drive myself to the market, and then sometimes drive to church."*

Nevertheless, they often relied on the principal driver whenever they planned for any family outings and trips. They also, however expressed that they would take over the role as principal driver if needed, especially when the partners were not available.

> *"Couple 3 Female: [Whenever we] go out together, he will drive."*

> *"Couple 10 Female: I will drive when my husband is not around. If my husband is not free [or unavailable], I will take the car and go."*

**Theme 2: Challenges to continue driving.** *Sub-theme 3*: *Money matters.* Participants reported a reduced need to drive after retirement from full-time employment or when all their children started working. For some, retirement ensued the need to be financially prudent when it involved driving.

> *"Couple 9 Female: Frequency of driving of course it is much reduced, because now I am a retiree. . ."*

> *"Couple 3 Male: The petrol is so expensive now, RM2.30 per litre."*

*Sub-theme 4*: *Physical and physiological changes within me.* Participants mentioned that they were very conscious about the physical changes and their driving abilities as they age.

> *"Couple 5 Male: I will have to be conscious of the symptoms or signs as I am growing old."*

> *"Couple 8 Male: When you reach a certain age, you will find it a bit difficult to drive."*

Many foresee that they may face some inevitable degenerative changes due to ageing that will affect their ability to drive safely and longer, including having one or more comorbidities, suffering from vision and cognitive impairments, reduced physical strength and flexibility, and poor psychomotor ability.

> *"Couple 6 Female: Maybe, if I had a stroke, then I will not drive."*

> *"Couple 8 Female: Maybe poor vision and poor cognitive ability. If these kinds of conditions happen to me, I will definitely stop driving."*

> *"Couple 5 Male: Of course, as I age, my reflexes are not so good, becomes slower."*

Driving was also described as physically exhausting and tiring for some participants, especially when it involved long distance trips. And if they were to cease driving eventually, they would prefer a gradual transition, unless they were physically disabled.

> *"Couple 7 Male: Driving can be tiring; it needs a lot of energy."*

"*Couple 5 Male*: *If I am incapacitated due to injury. I will have to force myself to stop driving.*"

Apart from physical changes, psychological factors include low confidence, being fearful of traffic conditions or other road users, and concern about their safety as well as others.

"*Couple 10 Male*: *We know that we don't have the confidence to drive, better not to drive. With all the heavy traffics, it's terrible, and all the youngsters being reckless on the road, scares me.*"

"*Couple 8 Male*: *Of course, not only yourself, you also have to think of others. I always think of safety of others when I drive.*"

*Sub-theme 5*: *Advice from significant others*. Participants expressed that they often received advice from their spouse on their driving. These could come in the form of guidance and sometimes, comment from their partner.

"*Couple 3 Male*: *My wife did mention until recently I am driving too close to the dividers.*"

"*Couple 11 Female*: *My husband advised me to go for driving classes like a refresher course.*"

Their acceptance toward receiving advice from others varied. There were only a few who took the advice seriously, whilst the others did not.

"*Couple 3 Female*: *Sometimes, [I would take] his advice if I felt that his advice is good.*"

Amongst those who did not, they viewed themselves as safe and experienced drivers.

"*Couple 2 Male*: *I am still [capable] at driving and [I feel] it is still the same.*"

"*Couple 4 Male*: *I have been driving for long time.*"

*Sub-theme 6*: *Challenging driving environment*. Some of the common challenges that the participants reported during driving were related to the environment they were in, namely; poorly maintained road signage, road conditions, complex road networks or unfamiliar places, reckless road users, and vehicle condition, including the size of the car (too big) or car with mechanical problems.

"*Couple 1 Male*: . . .*but not in Malaysia, the signages are [often] covered by branches [obstacles].*"

"*Couple 10 Male*: . . .*the young drivers are reckless [when they drive] on the road.*"

**Theme 3: Our driving strategies to ensure continued driving.**   *Sub-theme 7*: *Modifying our driving habits*. Most of the current drivers admitted that they are unlikely to cease driving completely but will consider modifying their driving habits (e.g., drive locally or drive less frequent) if needed to continue driving safely.

"*Couple 2 Female*: *No matter what, [I] have to continue to drive, even for short distances. You cannot stop [driving] completely.*"

"*Couple 8 Female*: *Maybe I will drive less but not stop driving completely.*"

Even though these strategies contribute to overall reduction in their driving, they would rather choose to limit their driving than to cease driving completely. Their primary goal was mainly to continue driving for as long as possible to keep doing what they want to do.

"*Couple 11 Male*: *Without driving, I cannot move around. Driving is very important for me.*"

Despite their active driving routines, they also admitted that their driving habits had changed and that they were being more careful by planning their trips or practice defensive driving. Other examples include imposing driving limitations such as making fewer trips, driving shorter distances, driving at lower speed, and avoid driving in challenging situations such as at night, in the rain, and during peak hours.

"*Couple 5 Male*: *I will always do [practice] defensive driving, and so in a way I will have to avoid certain situations that will cause injury or even death to others.*"

"*Couple 7 Male*: *Nowadays, I am more cautious in driving. I always plan my trip [before I drive].*"

"*Couple 8 Female*: *Mostly short distances, but last time I drove far [and] if I can avoid driving at night, I will just don't.*"

"*Couple 11 Male*: *I will try to avoid rain. Certain time like late night, I don't drive. . .I try to avoid peak hours, because [of] traffic jams.*"

*Sub-theme 8*: *Sharing our driving responsibilities*. Sharing the driving responsibility was common amongst older couples who continued driving. There are two major factors which were taken into consideration namely, spouse's condition and nature of travel. Most of the time, they would share their driving if they were travelling long distances or when the partner was feeling tired, sleepy, or unwell.

"*Couple 8 Male*: *Sometimes we share, especially if we are travelling to far places, then we take turns.*"

"*Couple 3 Male*: *If it is a long journey, then yes, we take turns in driving.*"

"*Couple 1 Male*: *Sometimes, she will feel giddy [sic], so I will take over.*"

*Sub-theme 9*: *Spouse as a co-driver*. There was a mix of responses for having the spouse as a co-driver (co-pilot). Some couples were positive due the benefits of having a co-driver (co-pilot) beside them, especially if they were making a trip to a new place.

"*Couple 7 Female*: *If I am not very sure of the place, I prefer to have someone with me.*"

"*Couple 7 Male*: *I prefer somebody beside me, guiding me with the direction, so that I can move faster.*"

They also admitted that having a co-driver (co-pilot) will be good in later stages of their lives. Conversely, a few did not favour having a co-driver (co-pilot), as they felt that it was a distraction.

"*Couple 8 Male*: *It is a good idea when I reach the later stage of my life.*"

*Sub-theme 10*: *Public transportation*. Many participants had reservation when it came to travel by public transportation. They expressed the difficulty in reaching destination, long waiting time, limited accessibility as well as unfamiliarity with the transit systems.

"*Couple 7 Male*: *There was one time when I took the bus, I did not know when to get down. I had the trouble to tell the driver where I want him to stop.*"

Nevertheless, participants mentioned that they were willing to use public transportation to: [1] avoid traffic congestion; [2] as an alternative to driving if they were to travel somewhere far; and, [3] time-and effort saving, especially when it involved the search for a car parking space.

"*Couple 4 Male*: *If I am going to a congested area, I will take the public transport. Because it's terrible to get stuck in traffic jams.*"

"*Couple 4 Male*: *When you drive, you have to find a parking space and this is a hassle.*"

*Sub-theme 11*: *Mixed mode transportation*. Unlike previous studies whereby mobility amongst older adults was often described as mono-mode (e.g., drive a car or travel by public transport all the way to reach their destination) and there was no sign of interchange between private and public transportations, travelling using two different modes of transportation was a unique coping strategy identified amongst older couples in this study. There were two common ways how it was applied in their travel routine. Most of them drove their car to the nearby train station and then used public transport to reach their destination.

"*Couple 1 Male*: *Now, if we are going to University Malaya Medical Centre, we drive up to here, leave the car here, and take the bus or Light Rail Transit, preferably bus, because if Light Rail Transit, you will have to walk further to the hospital. If bus, it will drop you at the entrance.*"

Another way is to depend on their spouse to drive them to nearby train station and then travel by public transport to continue their journey.

"*Couple 9 Male*: *I usually ask my wife to drive me to the station and then, when coming back, she will fetch me home.*"

## Discussion

Driving transition can potentially impact the lives of older drivers and people within their social circles, especially their spouse or partner. We found that there was some form of association between older married couple's engagement level in driving and their driving decisions, which influenced their subsequent driving transition outcomes. Although male partners were the principal drivers in this study, it is important to acknowledge the different roles held by female partners as reflected in the themes and sub-themes identified, including their roles as a co-driver or a driving partner to share the duty of driving. Ultimately, this study highlighted several key findings emphasising the concept of interdependency in older married couples and their engagement in driving transition, including passive strategies adopted by female partners, impact of personal factors (intrapersonal and interpersonal) on driving decisions, and mixed mode transportation as a coping strategy.

Whilst some of the factors and challenges (e.g., retirement, financial status, age-related changes, road system, and vehicle condition) identified in this study were previously reported in studies on older car drivers [2, 4], this study also noted how spouses influence driving self-regulation and how this decision influenced their driving roles (principal, alternate or former driver). It appears that one driver's engagement in driving is impacted by his/her partner's engagement simultaneously. This engagement may exist in various forms, including responsibility/duty (e.g., chauffeuring), opinion (e.g., advice, comment) or even just their presence (e.g., co-driver/navigator). Our findings were consistent with the conclusion of previous studies. For example, Rosenbloom and Herbel noted that male partners often took on the primary role of driving compared to their female counterparts [20]. In this study, majority of females expressed being dependent on their spouse for mobility and often drove only if needed. As such, most of the female partners in this study who were either an alternate driver or a former driver adopted passive strategies such as accompanying the principal driver (spouse) as a passenger or helping them with navigation as a co-driver (co-pilot), whilst assisting their spouse to maintain their driving competence. Previous studies also found that some older couples began their reliance on one another for transport, by serving as a co-driver (co-pilot) for each other emphasising the significance of female partners in sustaining mobility [21–23]. Nevertheless, in this study, the need for a co-driver (co-pilot) was perceived important only if they were to travel to unfamiliar places.

Whilst demographic and environmental factors (e.g., financial status and support systems) are important to ensure continued driving (Fig 1), personal factors are equally essential for older couples to be safe and mobile. These include driving abilities of the principal drivers (intrapersonal factor) and their willingness to bear the chauffeuring responsibility (interpersonal factor). In this study, mobility needs were maintained at satisfactory levels provided that at least one partner did not have any physical limitations affecting their ability to continue to drive safely. Most importantly, the driving partner must be willing to take charge of driving duty more than the other partner. A significant drawback of such strategy was that the driving partners that bear the responsibility may need to drive longer than they should. As such, this study suggests that self-regulation was less stressful and more adaptive amongst older married couples when both male and female partners continued driving. For instance, several couples adopted the strategy to share the duty of driving amongst themselves, given their nature of travel (long distances) and spouse condition (e.g. tired, sleepy, unwell). The advantage of such coping strategy for married couples was that it lessened the burden of the chauffeuring partner (often male partners) since both would take turns in driving.

In terms of alternative transportation, older married couples did not perceive public transportation as equal to private transportation, as they lamented the lack of convenience when using public transportation. However, older adults mentioned that they would use public transportation in certain situations, such as travelling to faraway places or to a congested area. Unique to this study, mixed mode transportation was a form of driving self-regulation for current car drivers. They reported to utilise two modes of transport by shifting between public and private transportations for a trip. This they felt was a smart and cost-effective coping strategy; whereby older adults were able to maintain their driving abilities and stay mobile.

## Implications and recommendations

In Malaysia, several national policies for older adults have been developed and implemented (e.g., National Policy for Older Persons and Plan of Action for Older Persons in 2011, National Health Policy for Older Persons in 2008), however there is currently no licensing system and medical assessment in place to ensure that older adults are medically fit and able to drive safely

[24]. Furthermore, many older adults do not perceive public transportation (e.g., bus, train, etc.) to be as convenient as private transportation, especially when there is no additional supports to assist them [4]. Given the rapid ageing population in Malaysia, it is timely and essential to conduct research to better understand and support the needs of growing older population. Ongoing gerontology and geriatric research is warranted to formulate, monitor, and evaluate the effectiveness of policies and programs implemented.

Whilst the key problem is to achieve the balance between safety and mobility in older drivers through policy and emerging technology, our study examined how married older adults help supported each other and adopted different driving and coping strategies in an effort to continue driving safely. The current study findings explored the impact of having at least one spouse who can drive on older couples' driving decisions and outcomes. We noted that the couple's engagement in driving may influence or constraint his/her partner's engagement in driving for transportation support [13, 16]. Whilst this is true, not many studies have examined the impact of living arrangement (living with spouse or other immediate family members) on driving self-regulation. Previous studies evidenced some form of interactions between living arrangement (multi-person household) and the incident of driving cessation [9, 25]. Traditionally, the family has been the main pillar of care and support for older people. Unlike Western community, Asians practice filial piety and often, adult children are the main providers of transport and financial support to the ageing parents [24]. However, demographic and social trends indicate more and more older adults will be living alone or with spouse only [2]. This suggests that future programs and interventions may need to consider the diversity of living arrangements as shaped by the presence of spouses, their roles, and contributions in driving transition and cessation planning, and the subsequent impact on driving decisions and outcomes. Given that driving ability is often linked to safety and mobility which affects well-being of the older populations [26,27], further research in this area is clearly warranted to better address this issue in the future.

## Strengths and limitations

To our best knowledge, this is the first study examining older married couples' perceptions of driving self-regulation and strategies adopted to maintain mobility. However, this need to be taken into consideration on the limitations of the study. Firstly, participants were urban dwellers; thus, the findings may be skewed to the themes reflecting perspectives of older couples from urban setting, limiting the generalisability to the general older population in Malaysia. Secondly, some of the responses obtained were retrospective reflections of their driving experiences underscoring the need for prospective studies (with follow-up interviews) to validate themes identified in this study. In term of analyses, there were only two former drivers and thus, and we were unable to provide robust evidence on the impact of having a spouse who ceased driving on older couples' driving decisions and outcomes. Lastly, all male partners were principal drivers and none of the female partners held active driving role at the time of data collection. Therefore, our attempt to further validate findings on the driving roles was not possible due to limited information. Overall, interpretation of themes and sub-themes should be cautious beyond our sample characteristics and future research would need to be conducted amongst a larger sample with greater focus on older couples with female principal drivers and former car drivers to validate current study findings.

## Conclusions

Current findings emphasised the importance of spousal influence (duty/responsibility, opinion, and presence) in driving self-regulation and the importance of couple-level engagement

for later life mobility status. We deduce that shared driving was an ideal coping strategy for older married couples, particularly amongst those who may not wish to cease driving. As evident in this study, the diverse roles played by spouses during the driving task and their contributions in driving transition were found to be influential, underscoring the need to recognise the importance of interdependency in couples and the subsequent impact on their driving decisions and outcomes. Therefore, future programs and interventions should consider the different roles and contributions of spouses in driving transition and cessation planning, especially for older couples to maintain their independence in mobility and access in safety.

## Supporting information

**S1 Checklist.**
(DOCX)

**S1 File. Guided questions.**
(DOCX)

**S1 Table. Demographic characteristics and driving history of participants who had participated in this study.**
(DOCX)

## Acknowledgments

The authors would like to thank late Dr Ngin Cin Khai for his contributions to this study. They would also like to thank School of Science in Monash University Malaysia for sponsoring tokens of appreciation for respondents.

## Author Contributions

**Conceptualization:** Boon Hong Ang, Jennifer Anne Oxley, Won Sun Chen, Shaun Wen Huey Lee.

**Data curation:** Boon Hong Ang.

**Formal analysis:** Boon Hong Ang.

**Investigation:** Boon Hong Ang.

**Methodology:** Boon Hong Ang, Jennifer Anne Oxley, Won Sun Chen, Shaun Wen Huey Lee.

**Project administration:** Boon Hong Ang.

**Resources:** Jennifer Anne Oxley, Won Sun Chen, Shaun Wen Huey Lee.

**Software:** Jennifer Anne Oxley, Won Sun Chen, Shaun Wen Huey Lee.

**Supervision:** Jennifer Anne Oxley, Won Sun Chen, Michelle Khai Khun Yap, Keang Peng Song, Shaun Wen Huey Lee.

**Validation:** Shaun Wen Huey Lee.

**Writing – original draft:** Boon Hong Ang.

**Writing – review & editing:** Jennifer Anne Oxley, Won Sun Chen, Michelle Khai Khun Yap, Keang Peng Song, Shaun Wen Huey Lee.

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
