## [Decision Letter · Decision Letter 0]

17 Feb 2020

PONE-D-19-33080

The influence of spouses and their driving roles in self-regulation: A qualitative exploration of driving reduction and cessation practices amongst married older adults

PLOS ONE

Dear Dr Lee,

Thank you for submitting your manuscript to PLOS ONE. After careful consideration, we feel that it has merit but does not fully meet PLOS ONE’s publication criteria as it currently stands. Therefore, we invite you to submit a revised version of the manuscript that addresses the points raised during the review process.

Please address the methodological issues raised by the reviewers.  Please also review for English language.  We recommend that you seek independent editorial help before submitting a revision. These services can be found on the web using search terms like “scientific editing service” or “manuscript editing service.”

Regards,

Rosemary Frey, PhD

Academic Editor Plos One

We would appreciate receiving your revised manuscript by 17 March 2020. To enhance the reproducibility of your results, we recommend that if applicable you deposit your laboratory protocols in protocols.io, where a protocol can be assigned its own identifier (DOI) such that it can be cited independently in the future. For instructions see: http://journals.plos.org/plosone/s/submission-guidelines#loc-laboratory-protocols

We look forward to receiving your revised manuscript.

Kind regards,

Rosemary Frey

Academic Editor

PLOS ONE

Journal Requirements:

2. Please provide additional details regarding participant consent. In the ethics statement in the Methods and online submission information, please ensure that you have specified whether consent was informed.

Reviewers' comments:

Reviewer's Responses to Questions

**Comments to the Author**

1. Is the manuscript technically sound, and do the data support the conclusions?

Reviewer #1: Yes

Reviewer #2: Partly

2. Has the statistical analysis been performed appropriately and rigorously? 

Reviewer #1: Yes

Reviewer #2: Yes

3. Have the authors made all data underlying the findings in their manuscript fully available?

Reviewer #1: Yes

Reviewer #2: Yes

4. Is the manuscript presented in an intelligible fashion and written in standard English?

Reviewer #1: Yes

Reviewer #2: No

5. Review Comments to the Author

Reviewer #1: This manuscript addressed an interesting and under investigated subject relating to how spouses impact the process of driving self-regulation in older adults. It presents novel findings that add to our understanding of this important age group. My reservations mainly relate to some of the interpretations made of the data and the clarity of the writing. I have made recommendations relating to these issues and should they be addressed I think the manuscript can make a valuable contribution. Below my comments are presented in three sections: ‘Clarifications required’ highlights issues that I am unclear about and need to be clarified before the implications of the research can fully be determined. ‘Recommended clarifications’ list changes that I think should be made which would made the communications clearer, but note if they are inconsistent with what the authors recommended then I am happy to be corrected. ‘Grammar’ lists the obvious errors that I spotted but I suspect there could be others. There is one instances were a citation needs to be provided by the authors to support the statement they are making and I have suggested two further references which I think will improve the manuscript.

Clarifications required

• L98 – how were the three different driving roles identified? If they were based on analysis of the current data then this analysis needs to be presented as preliminary analysis or part of the main analysis.

• Were all the interviews conducted by the same interviewer? This could impact of whether the interviews were conducted in a consistent manner.

• Where were the interviews conducted? Given the statements of some of the participants it would appear that they were conducted in the University but this should be stated explicitly.

• Where the interviews conducted in English or where they later translated into English? Some of the responses would indicate that they are translations e.g. L174 ‘I couldn’t think of any other support’ seems like an unlikely statement that someone would make. In instances where the translation or original speech is somewhat unusual it might be useful to include ‘sic’ to indicate that the transcription is replicated exactly from the recording e.g. in L283 ‘giddy’ appears unlikely.

• L179 ‘equally important’ needs to be supported with evidence from the data – in the current form it appears to be an unfounded assumption by the authors

• L202 -03 quote for C9F does not appear to relate to Money but rather an eagerness to avoid traffic jams and therefore needs to be replaced.

• L226 Not having the confidence to drive is different to Biological changes and either should be removed from this section or if possible developed as a separate theme – assuming that there are additional data to support it.

• L231 – sub-theme 5 is poorly tilted – consider something like ‘Advice from significant others’

• L240-41 – the comments from C2M and C4M require context and on their own do not contribute towards the Sub-theme 5.

• L243 – sub-theme 6 poorly tilted – the quotations provided relate to complexity of the driving environment not changes that have occurred to it.

• L262 – using ‘love’ seems like over interpretation and is not supported by the data provided.

• L268 – would have been good to have seen a quote relating to reduction of driving at night, in rain or during peak hours since the quotes provided do not relate to these but would suspect that such quotes do exist.

• In the discussion would like to see comment on the fact that although males are the primary drivers it is important to acknowledge the role of the females as reflected in sub-themes 8 and 9.

• L357 – ‘… continued access in safety and mobility’ – this does not quite makes sense and should be reworded.

Recommended clarifications

• L66 – Important to state that clinicians should be/can be involved in the process e.g. (Puvanachandra, Kang, Kirwan, & Jeffrey, 2008)

• L72 Given that you are discussing married participants it is important to point out that married individuals are more likely to be drivers than non-married and this held for both males and females (Gormley & O'Neill, 2019)

• L112 – the value of gift voucher should be quoted in an international currency such as American dollars or Euros.

• L133 – include the SD of age.

• L340 use of the word ‘clear’ is over interpretation and should be ‘appears’

• L340 should read .. one driver’s engagement in driving is impacted by his/her partner’s ...

• L404/5 consider ‘However, the findings need to be interpreted in light of the following limitations.’

Grammar

* indicates an added letter ** added word

• L47-48 ‘In many older adults, they often..’ should be ‘Older adults often..’

• L78 – insert ‘the’ between ‘of’ and ‘current’

• L80 – think ‘programming’ should be programs

• L92-93 -whereby individuals* reduce_

• L96 – individuals*

• L100 – takes*

• L120 – remove ‘for’

• L122 Insert ‘themselves’ after ‘familiarised’

• L128 – evidence_

• L138 ‘which drive car almost every day’ should be ‘who drove a car almost every day’

• L277 ‘two major factors which** were ….’

• L277/8 ‘travel nature’ should be ‘nature of the travel’

• L339 have should be has

• L345 – a* majority of females

• L346 remove first ‘as’

• L361 ‘to continue to** drive safely’

• L398 think this line should end with ‘which’

• L407 ‘..limiting the generalisability to the** general

• L422 ‘…that shared* driving….

Citation required

• L49 – ability to drive safely begins to decline.

References

Gormley, M., & O'Neill, D. (2019). Driving as a Travel Option for Older Adults: Findings From the Irish Longitudinal Study on Aging. Front Psychol, 10, 1329. doi:10.3389/fpsyg.2019.01329

Puvanachandra, N., Kang, C. Y., Kirwan, J. F., & Jeffrey, M. N. (2008). How good are we at advising appropriate patients with glaucoma to inform the DVLA? A closed audit loop. Ophthalmic and Physiological Optics, 28(4), 313-316. doi:10.1111/j.1475-1313.2008.00574.x

Reviewer #2: the authors have tackled an interesting and important topic - the influence/role of spouses in decisions about driving retirement. Unfortunately I do not think that the manuscript in its current form is ready for publication. My main comment is that many of the findings are not novel, as they relate more to individual considerations (finances, "biological changes within me", public transportation as examples) that have been described elsewhere. Shortening the manuscript to focus solely on the novel findings related to spouses would be helpful.

Additional copy-editing and grammatical checks would be helpful, especially by a native English speaker in the US/Canada/UK/Australcia. Examples of odd phrases or typos include "people who cares for me" (subheading), or phrasing at line 69. I have noted a few additional examples below but not all of them.

1. Sample - when/why did you stop recruitment? Was it based on thematic saturation or an a priori number?

2. Additional description of Malaysia (roads, transportation options etc) would be helpful for international readers. What are licensing policies for older adults? How might your findings be similar or dfferent in other nations.

3. The first paragraph of the discussion could be softened - "we found that there was a clear connection between older married couple's engagement level in driving and their subsequent driving decisions..." - this sentence is awkward but also I had not seen that clear connection in your results.

4. The implications section - first sentence is very long.

6. PLOS authors have the option to publish the peer review history of their article (what does this mean?). If published, this will include your full peer review and any attached files.

Reviewer #1: Yes: Michael J. Gormley

Reviewer #2: No

---

## [Author Response · Author response to Decision Letter 0]

11 Mar 2020

Response to Reviewers 

Manuscript ID: PONE-D-19-33080

Comments to the Author

Reviewer 1

This manuscript addressed an interesting and under investigated subject relating to how spouses impact the process of driving self-regulation in older adults. It presents novel findings that add to our understanding of this important age group. My reservations mainly relate to some of the interpretations made of the data and the clarity of the writing. I have made recommendations relating to these issues and should they be addressed I think the manuscript can make a valuable contribution. Below my comments are presented in three sections: ‘Clarifications required’ highlights issues that I am unclear about and need to be clarified before the implications of the research can fully be determined. ‘Recommended clarifications’ list changes that I think should be made which would made the communications clearer, but note if they are inconsistent with what the authors recommended then I am happy to be corrected. ‘Grammar’ lists the obvious errors that I spotted but I suspect there could be others. There is one instances were a citation needs to be provided by the authors to support the statement they are making and I have suggested two further references which I think will improve the manuscript.

Clarifications required

1. L98 How were the three different driving roles identified? If they were based on analysis of the current data then this analysis needs to be presented as preliminary analysis or part of the main analysis.

Response: The necessary changes have been included in the revised manuscript. Additional statement and supporting description can be found in the materials and methods under participants and definitions. Relevant explanation and changes made are as follow:

Explanation: The driving roles were identified through thematic analysis performed for this study and results can be found in the results section under thematic analyses (Theme 1: Our roles in driving).

Materials and Methods

Participants and definitions

“As we aimed to explore the role of spouse in driving self-regulation, we examined three different driving roles that were identified through thematic analyses performed for this study based on the participants’ responses (See Theme1 in the results), defined as follow:…”

2. Were all the interviews conducted by the same interviewer? This could impact of whether the interviews were conducted in a consistent manner.

Response: The necessary changes have been included in the revised manuscript. Additional statement and supporting description can be found in the materials and methods under data collection and instruments. Changes made are as follow:

Materials and Methods

Data collection and instruments

“To ensure consistency in data collection, the first author conducted all the interviews.”

3. Where were the interviews conducted? Given the statements of some of the participants it would appear that they were conducted in the University but this should be stated explicitly.

Response: The necessary changes have been included in the revised manuscript. Additional statement and supporting description can be found in the materials and methods. Changes made are as follow:

Materials and Methods

Data collection and instruments

“Interviews were conducted at a time and location (e.g. a closed room or a quite place at home, organisation, or medical centre) of convenience for the participants.”

4. Where the interviews conducted in English or where they later translated into English? Some of the responses would indicate that they are translations e.g. L174 ‘I couldn’t think of any other support’ seems like an unlikely statement that someone would make. In instances where the translation or original speech is somewhat unusual it might be useful to include ‘sic’ to indicate that the transcription is replicated exactly from the recording e.g. in L283 ‘giddy’ appears unlikely.

Response: The necessary changes have been included in the revised manuscript. For clarity, additional statement and supporting description can be found in the materials and methods under data analyses. Several changes were made to the quotes in the results under thematic analyses. Relevant explanation and changes made are as follow:

Explanation: All the interviews were conducted in English and there were no translation involved in this study. It is also important to note that English may not be their first language for most of the couples interviewed in this study.

Materials and Methods

Data analyses

“All the quotes included in the text were transcribed exactly from the original recordings. In the event of absence of clarity, additional descriptions were incorporated in parentheses to aid in understanding or [sic] was indicated where necessary.”

Results

Thematic analyses

“Couple 1 Female: If I stopped driving, I will depend on my husband.”

“Couple 1 Male: Sometimes, she will feel giddy [sic], so I will take over.”

5. L179 ‘equally important’ needs to be supported with evidence from the data, in the current form it appears to be an unfounded assumption by the authors

Response: The necessary changes have been included in the revised manuscript. Revised statement can be found in the results under thematic analyses. Changes made are as follow:

Results

Thematic analyses

“Just like principal drivers, alternate drivers held an important driving role in their early phase of driving, to support the family needs, including chauffeuring the children to school related activities or visiting their grandparents.”

6. L202 03 quote for C9F does not appear to relate to Money but rather an eagerness to avoid traffic jams and therefore needs to be replaced.

Response: The necessary changes have been included in the revised manuscript. Changes were made to the quotes in the results under thematic analyses. Relevant explanation and changes made are as follow:

Explanation: The quote was retained mainly to highlight her first response for reduction in driving frequency was due to retirement from a job. The latter quote was removed as it does not fit the current sub-theme.

Results

Thematic analyses

“Couple 9 Female: Frequency of driving of course it is much reduced, because now I am a retiree...”

7. L226 Not having the confidence to drive is different to Biological changes and either should be removed from this section or if possible developed as a separate theme assuming that there are additional data to support it.

Response: The necessary changes have been included in the revised manuscript. Revised sub-themes can be found in the results under thematic analyses. Changes made are as follow:

Results

Thematic analyses

“Sub-theme 4: Physical and physiological changes within me”

8. L231 Sub-theme 5 is poorly tilted, consider something like ‘Advice from significant others’

Response: The necessary changes have been included in the revised manuscript. Revised sub-theme can be found in the results under thematic analyses. Changes made are as follow:

Results

Thematic analyses

“Sub-theme 5: Advice from significant others”

9. L240-41 The comments from C2M and C4M require context and on their own do not contribute towards the Sub-theme 5.

Response: The necessary changes have been included in the revised manuscript. Supporting description can be found in the results under thematic analyses. Relevant explanation and changes made are as follow:

Explanation: The quotes were retained to support the last statement of the sub-theme. The position of the last statement and the two quotes were changed in the revised manuscript for clarity.

Results

Thematic analyses

“Amongst those who did not, they viewed themselves as safe and experienced drivers.”

“Couple 2 Male: I am still [capable] at driving and [I feel] it is still the same.”

“Couple 4 Male: I have been driving for long time.”

10. L243 Sub-theme 6 poorly tilted, the quotations provided relate to complexity of the driving environment not changes that have occurred to it.

Response: The necessary changes have been included in the revised manuscript. Revised sub-theme can be found in the results under thematic analyses. Changes made are as follow:

Results

Thematic analyses

“Sub-theme 6: Challenging driving environment”

11. L262 Using ‘love’ seems like over interpretation and is not supported by the data provided.

Response: The necessary changes have been included in the revised manuscript. Revised statement can be found in the results under thematic analyses. Changes made are as follow:

Results

Thematic analyses

“Their primary goal was mainly to continue driving for as long as possible to keep doing what they want to do.”

12. L268 Would have been good to have seen a quote relating to reduction of driving at night, in rain or during peak hours since the quotes provided do not relate to these but would suspect that such quotes do exist.

Response: The necessary changes have been included in the revised manuscript. Revised statement and additional supporting quotes can be found in the results under thematic analyses. Changes made are as follow:

Results

Thematic analyses

“Despite their active driving routines, they also admitted that their driving habits had changed and that they were being more careful by planning their trips or practice defensive driving. Other examples include imposing driving limitations such as making fewer trips, driving shorter distances, driving at lower speed, and avoid driving in challenging situations such as at night, in the rain, and during peak hours.

“Couple 5 Male: I will always do [practice] defensive driving, and so in a way I will have to avoid certain situations that will cause injury or even death to others.”

“Couple 7 Male: Nowadays, I am more cautious in driving. I always plan my trip [before I drive].”

“Couple 8 Female: Mostly short distances, but last time I drove far [and] if I can avoid driving at night, I will just don’t.”

“Couple 11 Male: I will try to avoid rain. Certain time like late night, I don’t drive…I try to avoid peak hours, because [of] traffic jams.”

13. In the discussion would like to see comment on the fact that although males are the primary drivers it is important to acknowledge the role of the females as reflected in sub-themes 8 and 9.

Response: The necessary changes have been included in the revised manuscript. Additional statement and supporting description can be found in the discussion. Changes made are as follow:

Discussion

“Although male partners were the principal drivers in this study, it is important to acknowledge the different roles held by female partners as reflected in the themes and sub-themes identified, including their roles as a co-driver or a driving partner to share the duty of driving…most of the female partners in this study who were either an alternate driver or a former driver adopted passive strategies such as accompanying the principal driver (spouse) as a passenger or helping them with navigation as a co-driver (co-pilot), whilst assisting their spouse to maintain their driving competence. Previous studies also found that some older couples began their reliance on one another for transport, by serving as a co-driver (co-pilot) for each other emphasising the significance of female partners in sustaining mobility (Kostyniuk and Shope, 1998; Freund & Szinovacz, 2002; Bonham et al., 2004)…this study suggests that self-regulation was less stressful and more adaptive amongst older married couples when both male and female partners continued driving. For instance, several couples adopted the strategy to share the duty of driving amongst themselves, given their nature of travel (long distances) and spouse condition (e.g. tired, sleepy, unwell). The advantage of such coping strategy for married couples was that it lessened the burden of the chauffeuring partner (often male partners) since both would take turns in driving.”

14. L357 ‘… continued access in safety and mobility’, this does not quite makes sense and should be reworded.

Response: The necessary changes have been included in the revised manuscript. Revised statement can be found in the discussion. Changes made are as follow:

Discussion

“Whilst demographic and environmental factors (e.g., financial status and support systems) are important to ensure continued driving (Fig 1), personal factors are equally essential for older couples to be safe and mobile.”

Recommended clarifications

15. L66 Important to state that clinicians should be/can be involved in the process e.g. (Puvanachandra, Kang, Kirwan, & Jeffrey, 2008)

Response: The necessary changes have been included in the revised manuscript. An additional reference can be found in the introduction and references. Changes made are as follow:

Introduction

“In most individuals, the process of transitioning from driving to driving cessation is not an individual experience, but often involves a wider social network, including their spouse, children, friends, and even professionals such as clinicians (Kostyniuk & Shope, 2003; Puvanachandra et al., 2008).”

References

Puvanachandra N, Kang CY, Kirwan JF, Jeffrey MN. How good are we at advising appropriate patients with glaucoma to inform the DVLA? A closed audit loop. Ophthalmic and Physiological Optics. 2008; 28(4), 313-316. DOI:10.1111/j.1475-1313.2008.00574.x

16. L72 Given that you are discussing married participants it is important to point out that married individuals are more likely to be drivers than non-married and this held for both males and females (Gormley & O'Neill, 2019.

Response: The necessary changes have been included in the revised manuscript. Additional statement and supporting reference can be found in the introduction and references. Changes made are as follow:

Introduction

“According to Gormley and O’Neill (2019), both married individuals were more likely to be active drivers.”

References

Gormley M, O'Neill, D. Driving as a Travel Option for Older Adults: Findings From the Irish Longitudinal Study on Aging. Front Psychol. 2009; 10, 1329. DOI: 10.3389/fpsyg.2019.01329

17. L112 The value of gift voucher should be quoted in an international currency such as American dollars or Euros.

Response: The necessary changes have been included in the revised manuscript. Revised statement can be found in the materials and methods under data collection and instruments. Changes made are as follow:

Materials and Methods

Data collection and instruments

“Couples who consented (written consent) to participate were interviewed separately and received a gift voucher worth RM 50 (approximately USD 11.96) each.”

18. L133 Include the SD of age.

Response: The necessary changes have been included in the revised manuscript. Revised statement can be found in the results under participant characteristics. Relevant explanation and changes made are as follow:

Explanation: The distribution of age was skewed, thus, non-parametric approach (median, range) was adopted to describe age of participants. Regardless, range of age was included in the revised manuscript to provide a clearer picture of age distribution of study sample.

Results

Participant characteristics

“A total of 11 couples were interviewed with 20 current car drivers and two former car drivers with a median of 68 years old (60-79) (S1 Table).”

19. L340 Use of the word ‘clear’ is over interpretation and should be ‘appears’

Response: The necessary changes have been included in the revised manuscript. Revised statement can be found in the discussion. Changes made are as follow:

Discussion

“In this study, it appears that one driver’s engagement in driving is impacted by his/her partner’s engagement simultaneously.”

20. L340 Should read, one driver’s engagement in driving is impacted by his/her partner’s…

Response: The necessary changes have been included in the revised manuscript. Revised statement can be found in the discussion. Changes made are as follow:

Discussion

“In this study, it appears that one driver’s engagement in driving is impacted by his/her partner’s engagement simultaneously.”

21. L404/5 Consider ‘However, the findings need to be interpreted in light of the following limitations.’

Response: The necessary changes have been included in the revised manuscript. Revised statement can be found in the discussion under strengths and limitations. Changes made are as follow:

Discussion

“However, this need to be taken into consideration on the limitations of the study.”

22. Grammar

* indicates an added letter ** added word

• L47-48 ‘In many older adults, they often..’ should be ‘Older adults often..’

• L78 – insert ‘the’ between ‘of’ and ‘current’

• L80 – think ‘programming’ should be programs

• L92-93 -whereby individuals* reduce_

• L96 – individuals*

• L100 – takes*

• L120 – remove ‘for’

• L122 Insert ‘themselves’ after ‘familiarised’

• L128 – evidence_

• L138 ‘which drive car almost every day’ should be ‘who drove a car almost every day’

• L277 ‘two major factors which** were ….’

• L277/8 ‘travel nature’ should be ‘nature of the travel’

• L339 have should be has

• L345 – a* majority of females

• L346 remove first ‘as’

• L361 ‘to continue to** drive safely’

• L398 think this line should end with ‘which’

• L407 ‘..limiting the generalisability to the** general

• L422 ‘…that shared* driving….

Response: The necessary changes have been included in the revised manuscript.

Citation required

23. L49 Ability to drive safely begins to decline.

Response: The necessary changes have been included in the revised manuscript. Supporting references were included to support the statement in the introduction. Changes made are as follow:

Introduction

“However, as an individual begins to age, his/her skill and ability to drive safely begin to decline (aAng et al., 2019;Ang et al., 2017; bAng et al. 2019).”

References

aAng BH, Jennifer O, Chen WS, Lee SW. Factors and challenges of driving reduction and cessation: A systematic review and meta-synthesis of qualitative studies on self-regulation. Journal of Safety Research. 2019 Jun 1;69:101-8. DOI: 10.1016/j.jsr.2019.03.007.

Ang, BH, Chen, WS, & Lee, SWH. Global burden of road traffic accidents in older adults: a systematic review and meta-regression analysis. Archives of gerontology and geriatrics. 2017 May; 72: 32-38. DOI: 10.1016/j.archger.2017.05.004.

bAng BH, Lee SW, Oxley J, Yap KK, Song KP, Kamaruzzaman SB, Chin AV, Tan KM, Khor HM, Chen WS. Self-regulatory driving and riding practices amongst older adults in Malaysia. Transportation research part F: traffic psychology and behaviour. 2019 Apr 1;62:782-95. DOI: 10.1016/j.trf.2019.03.014.

Reviewer 2

The authors have tackled an interesting and important topic - the influence/role of spouses in decisions about driving retirement. Unfortunately I do not think that the manuscript in its current form is ready for publication. 

1. Sample, when/why did you stop recruitment? Was it based on thematic saturation or an a priori number?

Response: The necessary changes have been included in the revised manuscript. Additional statement and supporting description can be found in the materials and methods under data collection and instruments. Changes made are as follow:

Materials and Methods

Data collection and instruments

“Recruitment of participants was conducted until thematic saturation was achieved when no new themes identified in the subsequent interview.”

2. Additional description of Malaysia (roads, transportation options etc) would be helpful for international readers. What are licensing policies for older adults? How might your findings be similar or different in other nations.

Response: The necessary changes have been included in the revised manuscript. Additional statement and supporting description can be found in the introduction and discussion. Changes made are as follow:

Introduction

“Nevertheless, the impact of spousal influence on driving transition amongst older couples is largely understudied. Until recently, most of the available studies have explored spousal influence on life satisfaction and social engagement amongst older couples in developed countries with limited information from developing countries like Malaysia (Rotolo & Wilson, 2006; Curl et al., 2015; Schryer et al., 2017).”

Discussion

Implementations and recommendations

“In Malaysia, several national policies for older adults have been developed and implemented (e.g., National Policy for Older Persons and Plan of Action for Older Persons in 2011, National Health Policy for Older Persons in 2008), however there is currently no licensing system and medical assessment in place to ensure that older adults are medically fit and able to drive safely (Teh, 2017). Furthermore, many older adults do not perceive public transportation (e.g., bus, train, etc.) to be as convenient as private transportation, especially when there is no additional supports to assist them (bAng et al., 2019). Given the rapid ageing population in Malaysia, it is timely and essential to conduct research to better understand and support the needs of growing older population. Ongoing gerontology and geriatric research is warranted to formulate, monitor, and evaluate the effectiveness of policies and programs implemented.”

3.The first paragraph of the discussion could be softened, "We found that there was a clear connection between older married couple's engagement level in driving and their subsequent driving decisions..." - this sentence is awkward but also I had not seen that clear connection in your results.

Response: The necessary changes have been included in the revised manuscript. Revised statement can be found in the discussion. Changes made are as follow:

Discussion

“We found that there was some form of association between older married couple’s engagement level in driving and their driving decisions, which influenced their subsequent driving transition outcomes.”

4. The implications section, first sentence is very long.

Response: The necessary changes have been included in the revised manuscript. Revised statement can be found in the discussion under implications and recommendations. Changes made are as follow: 

Discussion

Implications and recommendations

“Whilst the key problem is to achieve the balance between safety and mobility in older drivers through policy and emerging technology, our study examined how married older adults help supported each other and adopted different driving and coping strategies in an effort to continue driving safely.”

5. Additional copy-editing and grammatical checks would be helpful, especially by a native English speaker in the US/Canada/UK/Australcia. Examples of odd phrases or typos include "people who cares for me" (subheading), or phrasing at line 69. I have noted a few additional examples below but not all of them.

Response: The necessary changes have been included in the revised manuscript. Revised sub-theme can be found in the results under thematic analyses. Changes made are as follow:

Results

Thematic analyses

“Sub-theme 5: Advice from significant others”

Introduction

“Nevertheless, the impact of spousal influence on driving transition amongst older couples is largely understudied.”

6. My main comment is that many of the findings are not novel, as they relate more to individual considerations (finances, "biological changes within me", public transportation as examples) that have been described elsewhere. Shortening the manuscript to focus solely on the novel findings related to spouses would be helpful.

Response: Relevant explanation and supporting statement are as follow:

Explanation: Thank you for the suggestion. For clarity, most of the responses may relate more to individual considerations and this is expected as couples were interviewed separately in this study. Considering the topic of interest revolves around driving and safe mobility, we are aware of the similarities with previous findings and this has been addressed in the discussion. As such, some of the quotes under Theme 2 (Challenges to continue driving) and Theme 3 (Our driving strategies to ensure continued driving) may reflect individual considerations. Having said that, we also found similarities in their responses which formed the direction of the thematic analyses to produce a meaningful connection to reflect spousal influence including, the different roles in driving (Theme 1: Our roles in driving), advice from spouse (Theme 2: Challenges to continue driving) and driving strategies (Theme 3: Our driving strategies to ensure continued driving), especially sub-themes 8,9, and 11.

Discussion

“Whilst some of the factors and challenges (e.g., retirement, financial status, age-related changes, road system, and vehicle condition) identified in this study were previously reported in studies on older car drivers (aAng et al., 2019; bAng et al., 2019), this study also noted how spouses influence driving self-regulation and how this decision influenced their driving roles (principal, alternate or former driver).”

---

## [Decision Letter · Decision Letter 1]

22 Apr 2020

The influence of spouses and their driving roles in self-regulation: A qualitative exploration of driving reduction and cessation practices amongst married older adults

PONE-D-19-33080R1

Dear Dr. Lee,

We are pleased to inform you that your manuscript has been judged scientifically suitable for publication and will be formally accepted for publication once it complies with all outstanding technical requirements.

With kind regards,

Rosemary Frey

Academic Editor

PLOS ONE

Additional Editor Comments (optional):

Reviewers' comments:

Reviewer's Responses to Questions

**Comments to the Author**

1. If the authors have adequately addressed your comments raised in a previous round of review and you feel that this manuscript is now acceptable for publication, you may indicate that here to bypass the “Comments to the Author” section, enter your conflict of interest statement in the “Confidential to Editor” section, and submit your "Accept" recommendation.

Reviewer #1: All comments have been addressed

Reviewer #2: All comments have been addressed

2. Is the manuscript technically sound, and do the data support the conclusions?

Reviewer #1: Yes

Reviewer #2: Yes

3. Has the statistical analysis been performed appropriately and rigorously? 

Reviewer #1: Yes

Reviewer #2: Yes

4. Have the authors made all data underlying the findings in their manuscript fully available?

Reviewer #1: Yes

Reviewer #2: No

5. Is the manuscript presented in an intelligible fashion and written in standard English?

Reviewer #1: Yes

Reviewer #2: Yes

6. Review Comments to the Author

Reviewer #1: (No Response)

Reviewer #2: The article still feels long to me, but I defer to the editor on this. The revision addressed my concerns.

7. PLOS authors have the option to publish the peer review history of their article (what does this mean?). If published, this will include your full peer review and any attached files.

Reviewer #1: Yes: Michael Gormley

Reviewer #2: No

---

## [Editor Report · Acceptance letter]

5 May 2020

PONE-D-19-33080R1 

The influence of spouses and their driving roles in self-regulation: A qualitative exploration of driving reduction and cessation practices amongst married older adults 

Dear Dr. Lee:

I am pleased to inform you that your manuscript has been deemed suitable for publication in PLOS ONE. Congratulations! Your manuscript is now with our production department. 

With kind regards,

on behalf of

Dr. Rosemary Frey 

Academic Editor

PLOS ONE